Kananura *et al. Health Research Policy and Systems* 2017, **15**(Suppl 2):107

Health Research Policy
and Systems

**RESEARCH**

# Participatory monitoring and evaluation approaches that influence decision-making: lessons from a maternal and newborn study in Eastern Uganda

Rornald Muhumuza Kananura[1,2]*, Elizabeth Ekirapa-Kiracho[1], Ligia Paina[3], Ahmed Bumba[4], Godfrey Mulekwa[5], Dinah Nakiganda-Busiku[6], Htet Nay Lin Oo[3], Suzanne Namusoke Kiwanuka[1], Asha George[3,7] and David H. Peters[3]

## Abstract

**Background:** The use of participatory monitoring and evaluation (M&E) approaches is important for guiding local decision-making, promoting the implementation of effective interventions and addressing emerging issues in the course of implementation. In this article, we explore how participatory M&E approaches helped to identify key design and implementation issues and how they influenced stakeholders' decision-making in eastern Uganda.

**Method:** The data for this paper is drawn from a retrospective reflection of various M&E approaches used in a maternal and newborn health project that was implemented in three districts in eastern Uganda. The methods included qualitative and quantitative M&E techniques such as key informant interviews, formal surveys and supportive supervision, as well as participatory approaches, notably participatory impact pathway analysis.

**Results:** At the design stage, the M&E approaches were useful for identifying key local problems and feasible local solutions and informing the activities that were subsequently implemented. During the implementation phase, the M&E approaches provided evidence that informed decision-making and helped identify emerging issues, such as weak implementation by some village health teams, health facility constraints such as poor use of standard guidelines, lack of placenta disposal pits, inadequate fuel for the ambulance at some facilities, and poor care for low birth weight infants. Sharing this information with key stakeholders prompted them to take appropriate actions. For example, the sub-county leadership constructed placenta disposal pits, the district health officer provided fuel for ambulances, and health workers received refresher training and mentorship on how to care for newborns.

**Conclusion:** Diverse sources of information and perspectives can help researchers and decision-makers understand and adapt evidence to contexts for more effective interventions. Supporting districts to have crosscutting, routine information generating and sharing platforms that bring together stakeholders from different sectors is therefore crucial for the successful implementation of complex development interventions.

**Keywords:** Participatory monitoring and evaluation, Implementation research, Maternal and newborn health, Decision-making, Stakeholders

* Correspondence: mk.rornald@musph.ac.ug
[1]Department of Health Policy Planning and Management, Makerere University School of Public Health, Kampala, Uganda
[2]Department of Social Policy, London School of Economic and Political Science, London, United Kingdom
Full list of author information is available at the end of the article

# Background

The ever increasing demand for scarce resources has drawn more attention to the need to not only evaluate health programmes, but to also ensure that the results of these evaluations influence the implementation of programmes. The availability of accurate, timely and consistent data at the national and sub-national levels is assumed to be crucial for development programmes to effectively manage health systems, allocate resources according to need, and ensure accountability for delivering on health commitments [1–3]. A comprehensive monitoring and evaluation (M&E) system should enable programme implementers, decision-makers and budget planners to learn which strategies work and what needs to be improved so that resources can be better targeted towards saving lives [4]. Timely evidence from research during the course of implementation can inform and influence policy development, the identification of good practices and the development of sustainable health systems [4–6].

In contexts where maternal and newborn mortality is high, both demand and supply-side challenges exist side-by-side [2]. Comprehensive M&E systems are important for identifying the challenges that can eventually be mitigated. For instance, providing appropriate maternity care is a complex process that involves a wide range of preventive, curative and emergency services as well as several different levels of care – from the community to the facility and beyond [2, 7]. At the household level, there is a need to recognise maternal and newborn danger signs by family members so that appropriate services can be sought [8, 9]. At community level, accessibility to information on maternal and newborn service, proximity to the health facility and access to transport contribute to the increased utilisation of services from skilled personnel. At the facility level, equipment, supplies and medicines must be available to enable the health provider to make the correct diagnosis, provide appropriate treatment and make timely decisions so as to save the life of the mother and her newborn [7, 8]. Addressing these barriers to access should be informed by periodic collection of data that tracks implementation changes and challenges, which can be shared regularly/systematically with community stakeholders (such as community health workers, known as village health teams (VHTs) in Uganda, and community local leaders), health service providers and decision-makers at district and national level. Supporting community actions that can lead to the desired changes requires M&E approaches that allow information gathering and sharing in participatory ways so as to influence decision-making and action by key community-level stakeholders [9]. Such participatory evaluation approaches have been defined as "*applied social research that involves a partnership between trained evaluation personnel and practice-based decision makers, organizational members with program responsibility or people with a vital interest in the program*" [10]. Weaver and Cousins [11] categorise participatory evaluation into practical participatory evaluation, which is more utilisation oriented and mainly focused on local problem solving, and transformative participatory evaluation, which is more emancipatory in nature with a strong empowerment component aimed at addressing existing inequities. What differentiates participatory M&E from other M&E approaches is that it is responsive to local needs [10, 11], since this provides an opportunity to local citizens to be able to generate local applicable ideas and resources that are sustainable within their local context [2, 8, 12, 13]. Application of participatory M&E facilitates translation of implementation findings for stakeholders, thus allowing them to gain a better understanding of the intervention and its possible effects [13, 14]. It also enhances their use of the evaluation findings through their participation in the implementation learning and assessment process [15]. In addition, the involvement of different stakeholders helps to uncover diverse views, which guides debate and better understanding of the issues that affect the communities [11, 15]. Furthermore, the application of participatory M&E approaches strengthens the skills of the people involved, thus enabling them to contribute more to the successful completion of the research project [10, 11]. This approach has been identified as one of the best ways of transferring knowledge and research skills to the implementers, allowing them to continuously take charge of new programmes/projects without relying too much on external skills [10, 11].

In addition, the approach also encourages fairness, since it provides an opportunity for different groups, such as the voiceless within the society/community with a stake in the implementation or research, to be represented [11]. As a result, this can inform the redesigning and improvement of programmes that do not reach their intended beneficiaries [16, 17].

Several authors have proposed theories that explain the mechanisms that underpin the activities and consequences of practical participatory evaluations. Smits and Champagne [18] emphasise the importance of four key concepts, namely interactive data generation, co-construction of knowledge, local context of action and instrumental use. On the other hand, Cousins and Chouinard [19], emphasise the evaluation context and decision/policy setting as factors that influence the participatory interactive processes positively or negatively. This interactive process eventually influences evaluation knowledge production and evaluation utilisation.

## The MANIFEST project

We used a range of participatory M&E approaches during the implementation of the Maternal and Neonatal Implementation for Equitable Systems (MANIFEST) project. MANIFEST had several key interventions, which were implemented using a participatory action research approach.

They included (1) community mobilisation and empowerment through the community health worker home visits, community dialogue meetings, radio talk shows and messages; (2) improvement of financial and geographical access to care by promoting savings for delivery care and organising local transport; and (3) health systems strengthening through training of health workers, mentorship, supportive supervision and capacity-building of leaders in management. These interventions were provided only in the intervention area except for the radio talk shows and messages, which were aired on radios with listenership in the control areas as well and support supervision, which was routinely provided by the district health team in both the control and intervention area. More details about the intervention are available in the MANIFEST study design paper [20].

The MANIFEST project had a multisectoral group of stakeholders who played different roles. The research team comprised of members from the district level (district health officers, and district reproductive health focal persons) and researchers from the Makerere University School of Public Health and Johns Hopkins University School of Public Health. The district health officers (AB, GM and DNB), Makerere University team (RMK, EEK and SNK) and Johns Hopkins University School of Public Health (LP, AG and DHP) were all responsible for designing the intervention. The Makerere University team was also responsible for building the capacity of the local implementers by providing technical support to the district and sub-county teams, who were the lead implementers. The Johns Hopkins University School of Public Health team provided general oversight for implementation of the project together with the Makerere University team. The sub-county and district level stakeholders comprised of the health workers, various community leaders and decision-makers (religious leaders, political leaders and technocrats). The health workers, facility and district managers were responsible for ensuring that quality services were provided, while the political and technical leaders at sub-county and district level were responsible for providing oversight and ensuring that required decisions on maternal and newborn health (MNH) were made during sub-county and district council meetings or other such fora.

The community level stakeholders included men and women of reproductive age, VHT members, savings group leaders and local transporters. The men and women of the community were important stakeholders, since they made decisions about seeking appropriate care for mothers and newborns and preparing for birth by ensuring that they had the financial resources required in addition to planning transport and purchasing other requirements needed for the mother and newborn. The VHT members were responsible for doing home visits and conducting community dialogues, which were community meetings established to discuss MNH issues. Saving group leaders and transporters provided relevant services that contributed to increasing access to cash and transport for MNH. The local transporters were chosen by the savings group and they were responsible for providing safe transport services to health facilities during antenatal care (ANC) and at the time of delivery. Prior to the implementation of the project, refresher trainings and orientation meetings were provided for all the local implementers in the project. This was done to enhance their capacity to play their expected roles, as explained above. Continuous technical support was also provided throughout the implementation of the project by the Makerere University team and respective local supervisors. Further details about the trainings performed are available in Ekirapa-Kiracho et al. [20].

During study implementation, the research findings were analysed, synthesised and shared on a quarterly basis with the different stakeholders in the intervention area. The quarterly assessments and feedbacks involved the health workers, district health management team, sub-county and district political and technical personnel, other implementing partners, and researchers. Whereas numerous papers have been written about outcomes of evaluation studies, much less attention has been paid to the evaluation processes themselves [21]. The purpose of this paper is to examine how the participatory M&E approaches were used to monitor implementation progress, identify challenges and influence decision-making by community and district level stakeholders.

## Methods
### Implementation study area and design
The MANIFEST intervention was implemented in three districts of eastern Uganda from 2013 to 2015. The estimated population of the three districts was 1,106,100 (Kamuli 500,200, Kibuku 209,000 and Pallisa 396,900) [22]. The three districts have 104 health facilities, 33 in Pallisa, 17 in Kibuku and 54 in Kamuli [22]. In these areas, only approximately 1 of 2 pregnant women attend four or more ANC visits or deliver in health facilities, which is less than the 60% and 75% ANC and facility delivery national wide average estimates, respectively [23, 24]. The MANIFEST baseline study estimated the neonatal mortality rate to be 34 per 1000 live births [25], compared to the 27 per 1000 live births national estimates [23, 24].

### Data collection and sources
The data for this paper is drawn from retrospective reflection of the various data collection sources that included document reviews, project implementation review meetings, focus group discussions, key informant interviews, health facility support supervisions and household surveys.

Details of how the implementation study data was collected are available in a study design paper [20].

## M&E approaches

Our motivation for using the participatory approaches was mainly pragmatic and political [11]. The pragmatic approach was aimed at promoting problem solving. We therefore encouraged the involvement and participation of local stakeholders in assessing progress with implementation, identifying key lessons and challenges, and subsequently suggesting suitable solutions to the challenges identified. In relation to the political aspects, our aim was to make sure that we gather the support of the community leaders (politicians), the implementing team (health workers, community development officers, implementing partners and community health workers) and the community, including marginalised populations such as adolescents and disabled persons. The project provided avenues for these stakeholders to be able to critically understand the health challenges at both health facility, community and individual level through providing evidence and allowing

interaction, which in turn motivated them to take an active role in providing solutions to the problems identified.

We collected data during the design stage at the beginning, during implementation and at the end of the intervention, and consistently involved stakeholders at national, district, sub-county and community (village) level during data collection and dissemination. Figure 1 provides a summary of the M&E data collection approaches and tools that were employed as well as the stakeholder engagements that were undertaken.

### Stakeholder involvement

At the planning and design stage, a planning meeting that involved the research team members, health providers, district leaders, sub-county leaders and community members was conducted in order to identify community conditions/problems that lead to underutilisation of maternal health services and contribute to maternal and newborn deaths. These planning meetings were facilitated by the Makerere University School of Public Health research team. During the planning meeting, the stakeholders were asked to discuss how to address the problems identified

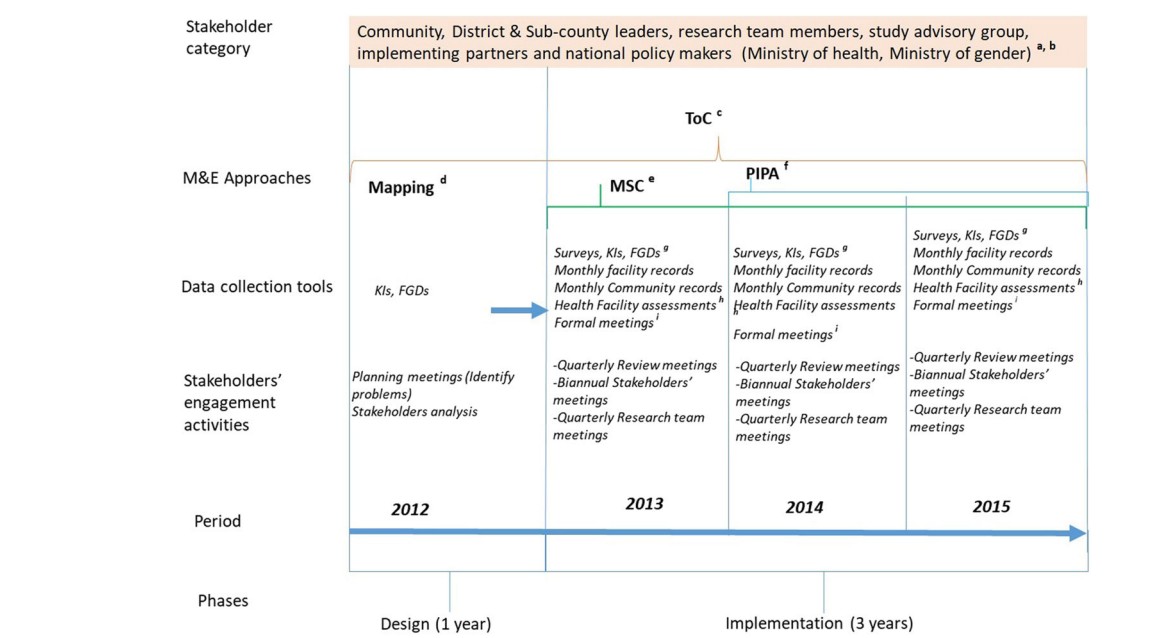

**Fig. 1** M&E tools, approaches, and activities used at different stages of programme design and implementation. **a** All stakeholders at national, district, sub-county and community (village) level as well as researchers were involved at planning and during the implementation phase. **b** National policy-makers were involved in biannual stakeholders' meetings conducted at national level and quarterly presentation at Ministry of Health reproductive health steering committee. **c** Theory of Change (ToC) was performed during design stage and was revised during the implementation through data collection and meeting with stakeholders at different levels. **d** Mapping was conducted during the design phase and included all stakeholders at national, district, sub-county and community (village) level. **e** The most significant change (MSC) approach was performed quarterly throughout the implementation period, with (**f**) participatory impact pathway analysis (PIPA) performed twice and involving only district stakeholders, sub-county stakeholders, implementing partners and researchers. This approach collected data through formal meetings. **g** Surveys, key informant interviews (KIs) and focus group discussions (FGDs) were performed at the beginning of implementation (baseline), quarterly/midterm and at the end of the implementation (end line). **h** Health facility assessments were performed at the beginning of implementation (baseline), quarterly/supportive supervision and at the end of the implementation (end line). **i** During formal meetings, for instance, quarterly review meetings, we collected data on significant changes and conducted PIPA workshops

using available resources and a given time frame. The involvement of the stakeholders at the planning stage provided a better understanding of the maternal and newborn problems and guided the selection of interventions that were implemented.

During the implementation phase, the stakeholders at the community and sub-county levels in the intervention areas were engaged in addition to the district level stakeholders. They were engaged through quarterly group meetings, which took place at sub-county and district level, quarterly support supervision visits to the health facilities, and quarterly group meetings with the VHTs and the communities (community dialogues). At district level, the meetings were chaired by the district health officer, who was responsible for mobilising all district stakeholders, including the implementing partners and donors such as UNICEF and USAID. At sub-county level, the meetings were chaired by the sub-county chief, who was also responsible for mobilising the sub-county implementation committee for the meeting. These stakeholder meetings have been described in Table 1.

During the stakeholders' meetings, the component lead, who was a member of the district health management team, presented results from study household surveys, health facility support supervision reports, key informant interviews and focus group discussions with the stakeholders/beneficiaries. Based on the presentations and discussions, appropriate actions were then taken by district planning leaders, health workers, health managers and the research team. The district biostatistician and the district health team were responsible for the analysis of routine data collected through the district health management information system, while the Makerere University research team was responsible for the analysis of data collected through additional surveys.

### Mapping and Theory of Change
This stage guided the team to map out the possible study outcomes, influential stakeholders to be targeted, partnerships to be identified, strategies for addressing community and health providers' behaviours, and inputs needed for the implementation of different strategies. This information

**Table 1** Description of stakeholder involvement

| Category | Participants | Aim |
|---|---|---|
| Sub-county quarterly review meetings facilitated by sub-county leaders | Sub-county implementation committee (technical, political and religious leaders at sub-county), district health team and the Makerere research team | Provide update on project implementation and uptake of interventions |
| | | Identify lessons learnt, implementation challenges and solutions |
| Quarterly research team meetings facilitated by both district and Makerere team | District health officers, district project focal persons and the Makerere research team | Provide update on project implementation and uptake of interventions |
| | | Identify lessons learnt, implementation challenges and solutions |
| District quarterly review meetings facilitated by district technical leaders | District implementation committee (technical, political and religious leaders at district level), district health team and the Makerere research team | Provide update on project implementation and uptake of interventions |
| | | Identify lessons learnt, implementation challenges and solutions |
| Support supervision led by district support supervision team and supported by the Makerere team | Health workers from intervention and control area | Monitor availability of MNH services |
| | | Identify gaps in MNH service delivery |
| | | Agree on action points with facility staff |
| | | Follow-up progress in addressing identified gaps |
| VHTs quarterly review meetings | All 1680 VHTs were involved in their respective sub-counties | Provide feedback to the VHTs about their performance and the community behavioural practices |
| | | Reinforce the knowledge and skills of VHTs |
| Community dialogue meetings led by VHTs and supervised by sub-county implementation committee | Community members | Discuss and promote local practices that influence MNH health positively and negatively |
| | | Discuss and discourage local practices that influence MNH health negatively |
| | | Encourage uptake of key intervention elements |

*MNH* maternal and newborn health, *VHTs* village health workers

was used to develop a Theory of Change. The Theory of Change enabled the research team members to clarify not only the ultimate outcomes and impacts they hoped to achieve, but also the avenues through which they expected to achieve them. This helped the research team and the local stakeholders build consensus on the implementation pathways. More details about the Theory of Change and how it was used are available in Paina et al. [26].

### Quantitative data collection

Quantitative information was collected through household surveys, health facility support supervision visits, health information utilisation data and reports from the community health workers. We conducted household surveys at baseline, midterm and endline so as to determine changes in the study outcomes, while we used Lot Quality Assurance Sampling (LQAS) techniques to conduct quarterly household surveys during the first 9 months of the study to monitor the uptake of key intervention elements. The main outcomes for LQAS household surveys were changes in facility deliveries, ANC attendance, birth preparedness practices, and knowledge of birth preparedness, pregnancy, labour and newborn danger signs. Every quarter, we randomly selected five villages as supervision areas in each district (supervision units), from which we randomly sampled 19 eligible households for assessment. A team of five district-based persons (government employees), who included the biostatistician and health management information system focal person, collected the data. Table 2 provides details of these data collection methods.

### Qualitative data collection

A team of trained research assistants with support from a qualitative research specialist from Makerere University collected qualitative data through focus group discussions, key informant interviews and quarterly review meetings at district and sub-county level. We conducted focus group discussions with men and women in rich and poor communities and in locations that were considered hard to reach and easily accessible. These areas were selected by members from the district health office [20]. The key informant interviews were conducted with community leaders, district health management team members and health providers [20]. More details about the data collection are described in Table 2.

### Most significant change

We used a modified version of the most significant change approach to help us track the most significant changes experienced by the health providers and the community during the implementation phase [27] (Fig. 1). We did this by collecting stories of change during focus group discussions with the community, key informant interviews with health providers and local leaders, and meetings (quarterly meetings, health workers symposia and research team meetings). The stories spanned across several domains that included quality of care provided at the health facilities, health workers' attitudes, changes in healthcare management/leadership skills and behavioural changes among mothers in terms of birth preparedness and newborn care. However, we did not rank these stories so as to identify the most significant change; rather, we considered all of them as stories of change since our aim was to capture perceptions of change from the stakeholders' perspective.

### Participatory impact pathway analysis (PIPA)

We used PIPA to identify key stakeholders involved in MNH. The PIPA workshop was conducted in the first

**Table 2** Description of data collection methods

| Data collection methods | Participants | Type of data |
| --- | --- | --- |
| Household surveys (baseline, midterm, endline) and quarterly monitoring surveys for the first three quarters of the intervention | Women and men of reproductive age | Participant demographics, birth preparedness practices, MNH service utilisation, newborn care practices, newborn death, saving practices, transport used to the health facility |
| Focus group discussions | Women and men of reproductive age | Perceived quality of MNH services, factors influencing MNH service utilisation and delivery, newborn care practices, saving practices, attendance of community dialogues and associated factors, access to transport services, birth preparedness, male involvement, perceptions about the MANIFEST intervention implementation |
| Key informant interviews | Health workers, local leaders and district health management team | |
| Supportive supervision | Facilities that provide MNH services | Availability of MNH services, availability of essential drugs, equipment and skilled health workers |
| Health facility records review | Facilities that provide MNH services | MNH service utilisation data, stillbirths, newborn deaths and maternal death |
| VHT monthly reports | VHTs from 840 villages in the intervention area | Monthly reports on newborn deaths, maternal deaths, women reached during home visits disaggregated by age |
| VHT surveys | VHTs | Knowledge about danger signs during pregnancy, delivery and postpartum, knowledge about the savings and transport component |

*MNH* maternal and newborn health, *VHTs* village health workers

and second year of implementation. Details about how it was conducted are available in Ekirapa-Kiracho et al. [28]. We used PIPA to analyse the type, role and strength of each stakeholder, as well as how they were connected with one another in the context of maternal and newborn services. This helped the project team to understand the actors in MNH, the resources that they possessed, as well as the power and influence that they had in promoting achievement of the project objectives.

## Results

In the subsequent sections, we present findings that illustrate how M&E information shared with each group of key stakeholders was used to identify challenges and linked to the decisions or actions taken.

### Community level

During the design phase of the programme we held focus group discussions and stakeholder meetings with local members of the communities. The purpose of these discussions were to identify local problems and feasible solutions as well as the existing local resources, including existing infrastructure and governance structures, human and financial resources. Through the discussions we were able to identify the problems that affect MNH services in three main areas, including birth preparedness, transport and quality of MNH care services in the health facilities. The problems related to birth preparedness included lack of awareness of its importance, negative cultural practices, men neglecting their roles, lack of knowledge about family planning, poor saving culture and poverty. The transport problems included absence of ambulances, long distances to health units, lack of appropriate transport vehicles and high transport fares. The quality of care was being compromised by frequent essential drug shortages, inadequate number of delivery beds, understaffing, poor health workers' attitudes, irregular support supervision, staff absenteeism, informal charges, and poor technical and managerial skills. This information was used to identify the interventions that were implemented. For instance, to address the challenge of low awareness about the importance of birth preparedness, home visits by community health workers were suggested and later included as one of the key interventions. To address poor managerial and technical skills, refresher training for health workers was proposed and provided as one of the interventions for health system strengthening.

The local resources identified included existing infrastructure and governance structures such as the sub-county committee, community development office, local transport associations, VHTs and savings groups. The sub-county committee was given the responsibility of supervising the quarterly community dialogues that were held at every village. The community development office was able to

provide technical support to the saving groups when we realised that most of them had managerial problems and lacked the basic documentation that was required for their efficient functionality.

During the implementation phase, we shared information about uptake of the intervention elements and progress with implementation of the intervention with the community level stakeholders. Table 3 provides a summary of key issues that were identified at the community level and shared with community stakeholders, as well as the actions that were recommended by these stakeholders.

Data from the household surveys provided information about the uptake of various aspects of the intervention. For example, in some of the hard-to-reach areas, newborn deaths were high and most of the women were delivering at home with assistance from traditional birth attendants. Data collected from community health workers also helped the research team and district health office capture the number of newborn deaths and maternal deaths more completely and accurately. Previously, the district only had data from the facility, which reflected a much smaller number of maternal and newborn deaths. The focus groups were used to explore the reasons behind these home deliveries and newborn/maternal deaths in more depth and to identify possible solutions that could be undertaken by community, facility or district level stakeholders. Table 4 provides a summary of the main factors contributing to maternal and newborn deaths and solutions that were proposed.

The main factors included delays in deciding to seek care and inadequate care at the health facilities, with delays in deciding to refer mothers at the health facilities. Some of the problems that had been identified during the problem identification phase were still present even at the design phase of the study. Their persistence during the intervention showed that more attention needed to be given to addressing them. These issues were then brought to the attention of local leaders, health providers, including VHTs, and district planners in the community. For example, through the community dialogues, we emphasised the importance of delivering in health facilities and preparing for birth by saving money so that transport could be availed in case a mother was referred to a more specialised facility. As a result, women started saving with the saving groups and some groups bought their own boda bodas, which they started using to transport the members of the groups at subsidised costs and sometimes for free. Initially, the community used to save mainly to meet their needs during festive seasons such as Christmas or for burial. However, this has now changed as a result of these community engagements, which have made the community realise the importance of birth preparedness and saving to meet their MNH needs (Additional file 1).

As alluded to earlier, we performed surveys with the VHTs to identify their knowledge about danger signs and

**Table 3** Community level information and actions taken

| Emerging issues | Data collection methods and avenues for information sharing | Actions suggested and taken |
|---|---|---|
| *Uptake of interventions by the community* | Data was collected through household surveys and shared during quarterly review meetings conducted at sub-county and district level | Conduct maternal and newborn audits at the community and health facilities to find out the reasons for the deaths |
| Some mothers still deliver at home and so maternal and newborn deaths reported in some communities | | |
| Mothers continue to bathe newborns immediately within 12 h after birth (86%) | | More health education about newborn care practices during home visits, community dialogues and at the health facility |
| Mothers continue to put local herbs on newborn cord (44%) | | |
| Poor attendance of community dialogues partly attributed to lack of involvement of local council leaders | Data was collected through key informant interviews and focus group discussions and shared during review meetings held at sub-county and district level | Sensitisation meetings held for local council leaders to inform them about their role in the study |
| *Factors influencing competence of VHTs in performing their duties* | | |
| VHTs lacked adequate knowledge about newborn danger signs (46%) | Data was collected through VHT surveys and shared with VHTs at VHT quarterly review meetings | Refresher training done during the quarterly group meeting and a change was noted (46–60%) |
| VHTs were not encouraging mothers to join saving groups and link up with transporters | Data was collected through VHT surveys and shared with VHTs at VHT quarterly review meetings | Refresher training of VHTs was done during quarterly group meeting and more information provided about transport and savings component; list of saving groups also given to VHTs |

*VHTs* village health workers

areas of weakness in conducting health education and referral during home visits. Results from the second monitoring data collection exercise (6 months after the intervention started), during which interviews were done with VHTs, revealed that only 46% knew at least three newborn danger signs. Furthermore, only 29% of the VHTs were knowledgeable about the transport and savings component of the intervention (Table 3). These results were shared with the VHTs during the quarterly group meetings and refresher training was provided in these weak areas. The VHTs also performed role-plays that reminded them of the information that they were to share with the rest of the community. We also used the quarterly group meeting as a method of obtaining feedback from the VHTs about how the health workers were responding to the clients. For example, early in the study, the VHTs reported that the health workers rejected some of the referrals made by them. This information was shared with the health workers, who explained their response by saying that sometimes the VHTs did not assess the patients well before referring them. For instance, some of them referred all women with big legs to the facility thinking they had oedema. This feedback was in turn given to the VHTs through quarterly meetings, who were then asked to ensure that they adequately assessed patients before referring them.

The PIPA workshops were used to identify stakeholders involved in MNH and their roles in improving access to MNH care. They enabled us to identify other implementing actors who could contribute to achieving the project objectives. For instance, default from payment was a challenge in the saving groups. During the workshop, it was recommended that the police could equip the community development officers with information about how to seek legal redress so that they are able to ensure that the saving groups can recover their money or protect it from being borrowed illegally. The PIPA workshop was also able to demonstrate to the community members that a multisectoral approach was required to increase access to care for mothers and newborns, since they were able to appreciate the role that women's husbands, VHTs, transporters, family members and health workers played in influencing where women delivered from. This increased their willingness to participate in activities that were geared at saving the lives of mothers and newborns.

### Health facility level

At facility level, the M&E data helped us to track MNH service availability and gaps. Table 5 provides a summary of issues that were identified and actions that were recommended.

The quarterly support supervision visits that involved both the research team and the district leaders helped the team to identify service delivery gaps, which were reported to the respective health facility teams and district health office for their action. For instance, in some facilities, health workers were not using partographs to monitor the labour progress and, in fact, some of the health workers did not know how to use the partograph.

**Table 4** Factors contributing to maternal and newborn deaths and solutions proposed

| Key issue identified | Solutions proposed |
|---|---|
| Delay in deciding to seek care for ANC and delivery<br><br>Delay in deciding to refer the mother to hospital<br><br>*"The first time she attended ANC, she was advised to go to the hospital. However, she never went because she thought using the local herbs would cure her. When the time for delivery reached, she went to [HC III] (immediately the labour started, 8:00 am). When the facility staff failed, they referred her to the hospital at 11:00 pm (at night). The hospital opted for a caesarean. After the operation, she bled too much, and this resulted in her death. Fortunately, the baby survived"* Deceased's sister | Religious leaders, community health workers and local leaders to continue participating in sensitising their communities on the importance of accessing maternal health services from health facilities during home visits and community dialogues; this was proposed during sub-county second quarter meeting during the first year and it was done<br><br>Strengthen monitoring of women in labour using partographs through mentorship and support supervision so that referrals are not delayed; this was proposed by the district health management team in second quarterly meeting during the first year of implementation, Makerere University then trained district mentors who conducted subsequent mentorship and support supervision to health workers |
| Poor health worker attitudes<br><br>*"I went to the facility when my pregnancy was 2 months but was denied access to services because I had not gone with my husband. I again went there when it was 6 months and the same happened. …. I tried to explain to the health worker, but she could not listen to me. When the time for delivery reached, I decided to deliver from home because I feared to go back to the facility. Two days after delivery, my child died"* Mother 35 y, gravid 5 and above | Health workers advised to relax the policy of only working on women who attended ANC with their partners; this was suggested by the district local leader (Local Councillor V); the decision was welcomed since the main aim is to make sure women reach health facilities regardless of whom they go with |
| Delay in deciding to refer the mother<br><br>Lack of immediate transport for referral<br><br>*"I reached the [HC] at 2:00 pm but was referred to hospital at 3:00 am… the health workers found that they could not manage me and I was referred to the [regional referral hospital]. Unfortunately, the driver for the ambulance was not around, and the vehicle was got at 4:00 am…. When I reached [regional hospital], a decision was made to do a caesarean. Unfortunately, the baby died immediately after delivery"* Mother, 29 y, gravid 5 and above | Strengthen monitoring of women in labour using partographs through mentorship and support supervision so that referrals are not delayed; this was suggested by the district health management team in second quarterly meeting during the first year of implementation, Makerere University then trained district mentors who conducted subsequent mentorship and support supervision to health workers<br><br>District health office to work with CAO to make sure the ambulance driver and fuel are always available to ease referral; this was suggested by CAO and a budget line to avail money for the driver and ambulance was allocated immediately during the meeting |
| Lack of health worker skills in managing obstructed labour<br><br>*"I attended ANC four times at [HC III]. During delivery, the baby's head came out, but other parts could not come out. I tried to push but it could not come out. Unfortunately, it died before even coming out. I think it was too big"* Mother 24 y, gravid 3 | Obstetricians and gynaecologists to continue mentoring midwives on how to handle complications during delivery through mentorship; this was suggested by the district health officer and Makerere University School of Public Health agreed to take-up this role of training mentors who will be responsible for scaling up the skills in all health facilities within the district |

*ANC* antenatal care, *CAO* chief administrative officer, *HC* health centre, *y* years

However, during support supervision by the district support supervision team, health workers were reminded about the importance of using partographs and provided with refresher training on the use of partographs [29] (Table 5).

In addition, some health facilities were found to have no essential birth items and equipment. In some cases, the facilities had this equipment in their stores, but they were not aware of their availability. For example, at least five lower level health facilities in Kibuku and the health district store in Pallisa had manual vacuum aspiration sets that they were able to put to use. The support supervision reports were discussed in the district quarterly review meetings, which tasked the health facility and district teams to find alternative means of addressing these problems.

The periodic household surveys helped us identify newborn care gaps. For example, the midterm survey results indicated that few mothers with low birth weight infants received information on how to care for small infants (36.8%), and only 5.3% received kangaroo mother

care (midterm 2014 report). These findings were shared with the health workers during support supervision and mentorship visits and the district health office during quarterly meetings. As a result, the district health offices and the Makerere research team decided to support the health facilities to put in place strategies that may strengthen the newborn care services focusing on care for low birth weight infants as suggested by the health workers. Hospitals and health centres in each of the districts set up newborn care corners. In addition, the study team put more emphasis on building skills related to the management of newborn infants by adding a paediatrician to the mentorship team and provided more skills-building sessions on low birth weight infant screening and management of pre-term infants with a focus on newborn resuscitation skills and kangaroo mother care.

### Sub-county and district level
During the sub-county and community quarterly review meetings, the research team, district health team and health workers provided an update about the uptake of various

**Table 5** Health facility level information and actions taken

| Emerging issues | Data collection methods and avenues for information sharing | Actions suggested and taken |
|---|---|---|
| *Monitoring of women in labour* | | |
| Limited use of partographs to monitor the progress of labour | Information collected through supportive supervision visits and shared through district review meetings | Training of the health workers on the use of the partograph through mentorship programme and support supervision; training was done by Makerere University and health facilities started budgeting for the printing partographs using their primary health centre fund |
| Maternal and newborn death high in some health facilities | Data was collected through records review/supportive supervision and shared during quarterly review meetings | Maternal and newborn death audits were recommended; the District reproductive health focal person found that, in one hospital, the nurses did not know how to resuscitate newborns, so it was suggested that this nurse receives a training, which was done by attaching a district mentor at this facility; in another facility, unnecessary augmentation of labour was being performed, leading to foetal distress and stillbirths, so the midwife was given guidance by district mentor about when to augment labour |
| *Care for newborns* | | |
| Poor care of small infants – neonatal resuscitation and using Kangaroo Mother Care | Data was collected through midterm household surveys and shared in the second quarterly review meetings during second year of implementation | District health officers requested Makerere University to design a mentorship programme focusing on caring for small infants; Makerere University School of Public health mentored the district mentors who in turn scaled-up the skills to other facilities |
| | | District officer in charge of paediatrics proposed putting in place a newborn care corners started at the health facilities; Makerere University School of Public Health brought in a paediatrician on the mentorship team so as to improve newborn care |
| *Resources for providing maternal and newborn services* | | |
| Stock-out of maternal and newborn essential drugs and supplies | Information collected through supportive supervision visits and shared through district review meetings | Training the health facility managers on proper drug requisitioning during the certificate course on management by Makerere University School of Public Health; however, in some cases, a persistent drug stock-out was brought about by the delay in the delivery of supplies by National Medical Stores – a body that is responsible for the distribution of drugs in all health facilities; nevertheless, facilities that had excess shared with facilities that had inadequate amounts |
| Four health facilities did not have a placenta pit for disposal of placentas | | The sub-county leadership was informed at the sub-county review meeting and they availed funds to construct the placenta pits; the placenta pits were built in all facilities with the support from the sub-county |
| Some hospitals and health centre IV did not have an ambulance | Data collected through health facility assessment and review meetings | Political leaders to lobby politicians and other stakeholders to buy ambulances; members of parliament in Pallisa district bought four motorised ambulances |
| | | One sub-county bought a motorcycle ambulance |
| | | Fundraising was done and 10 trailers for motorcycle ambulances were purchased |
| Ambulances have mechanical problems and cannot transport women | | Medical superintendent for the hospital was asked to ensure funds allocated for repair of the ambulance during district review meeting and this was done (Pallisa district) |
| No fuel for the hospital ambulance | | The district health officers availed money for fuel for the ambulance from the budget line at district level (Kibuku district) |

aspects of the intervention (home visits by VHTs, maternal and newborn care practices, attendance of community dialogues, formation of saving groups and linkages with transporters), gaps in health service delivery and challenges faced (Tables 3 and 5). The sharing of these findings enabled the community leaders and decision-makers to learn about local conditions and problems affecting the communities and they were able to take actions to respond to those problems. For example, during supportive supervision visits, we were able to find out that some facilities did not have placenta pits for placenta disposal, essential drugs and supplies, electricity or fuel for the ambulance. The sharing of this information in the review meetings at sub-county and district level prompted the leaders and decision-makers to take the required action. The district and sub-county offices availed the funds required for construction of the placenta pits, repairing of health facilities and ambulances as well as fuel provision (Table 5). Information sharing also helped district officials to identify additional resources and partnerships that could be made available. For example, the Member of Parliament in Pallisa district bought motorised ambulances to ease the transport problem and the sub-county leaders considered procurement of tricycles and facilitation of community health workers in their budget planning (Table 5).

The involvement of the health district leaders, health facility managers and sub-county leadership team in planning and M&E also strengthened their capacity in the use of data for advocacy, planning and decision-making. As a result, some district health offices have learnt the importance of participatory planning, M&E as a tool for information sharing, advocacy and resource allocation. For instance, Kibuku district now uses the health information data to determine the facilities that need to be expanded or that require more staffing (Additional file 2). Some of the health facility managers also reported that they now hold more effective meetings with clearly spelt out objectives and action plans. In addition, they are able to garner more support from stakeholders who influence decision-making about allocation of resources. As a result, one of the health facilities was able to receive funds for the renovation of its theatre (Additional file 3).

## Discussion

This paper describes the participatory M&E methodologies and tools used to identify key implementation issues and solve problems and how they influenced decision-making. Some of the key components noted in theories about practical participatory M&E, such as interactive data generation, local contextual action and local contextual factors, were also noted to have played key roles in data generation and influencing decision-making in this paper [11, 19]. Use of a combination of M&E approaches and tools had several benefits. Firstly, they allowed triangulation of data from various sources leading to more complete reporting and a better understanding of some of the issues noted. They also enhanced interactive data generation, which enabled the study implementation team to get perspectives from different stakeholders and provided a comprehensive picture of how different factors and actors were interacting to influence MNH outcomes. For example, several newborn deaths were reported in the intervention area. Hence, it was important for the district health management team, health workers and other key stakeholders to understand the circumstances that led to the deaths, so that measures could be put in place to stop similar occurrences. The qualitative interviews that were done with women who had lost their infants therefore aided in the identification of the factors that contributed to these deaths. The district and sub-county level stakeholders were then able to take actions to solve some of the problems identified. Other researchers have also indicated the importance of combining quantitative assessments of pre-specified mediating variables with qualitative investigation of participant responses in testing and redefining the causal intervention assumptions [8, 16, 30, 31].

Secondly, combining qualitative and quantitative data collection methods from several sources and discussing these results with a diverse group of stakeholders during the quarterly review meetings also allowed co-construction of knowledge and the identification of unanticipated pathways, as well as in-depth exploration of pathways which are too complex to be captured using one method [30]. For instance, to encourage male involvement, facilities prioritise women who come with their partners and sometimes decline to work on women who do not during ANC. However, we noted that this becomes a barrier to seeking formal delivery care services for women who have no partners. These women feel discriminated against and decide to shun all the facility services, as described by a woman who gave birth at home and later lost her newborn. During the meeting, these findings on the community stories were shared and the district health offices and leaders agreed that the policy of not attending women who have not come with their husbands should be abolished.

Thirdly, the frequent interactive monitoring of the implementation allowed us to identify gaps in implementation and to identify practical solutions that could be implemented by those who were responsible for improving service delivery. During supportive supervision, we realised that, whereas some of the implementation was constrained by factors that were beyond the control of the facilities or health workers, such as inadequate essential equipment and supplies and skilled motivated health workers [32–34], in several cases there was something that the health workers could do. For example, some newborns were dying without being resuscitated and partographs were not being filled in some facilities simply because the health workers lacked the skills to do so.

Availing information on these health facility gaps, and emphasising the fact that the health workers could change this situation if supported, encouraged the facility and district management to take action whenever there was a problem that they could solve. Other researchers have indicated the importance of engaging clinical and management staff in the discussion of implementation barriers and facilitators [33–35]. However, for this monitoring to lead to improvement in service delivery, the gaps must be clearly specified, actions that are to be taken to mitigate them must be identified and persons responsible indicated, and follow-up must be carried out to ensure that the required action was taken, otherwise the problems simply continue to persist.

Finally, the participatory M&E methods facilitated interactive processes that promoted interaction and dialogue between the stakeholders. We realised that the dialogue enhanced the ability of stakeholders to hold each other accountable, which was an unanticipated positive outcome. Other studies have also emphasised the need for information sharing with stakeholders at each stage of implementation [5, 8], which strengthens appropriate decision-making, advocacy and resource sharing [12, 36, 37]. However, we noted that, in the absence of the research team, things tended to slide back to the status quo, implying that strengthening of such accountability processes requires time and local champions before it can become entrenched into local systems.

In spite of the above benefits, we note that there were several challenges that may hinder instrumental use of findings from participatory M&E approaches and may also limit the decision-making, especially in low- and middle-income countries. One of the major challenges was related to local contextual factors. Often, the available resources were not adequate for taking the required actions. One of the key weaknesses noted was in the referral system, which needed a comprehensive set of communication and transport facilities at the community and facility levels. Although some progress was made through the purchase of a motorcycle ambulance and trailers as well as the purchase of motorised ambulances by politicians, these were not enough. Persistent inability to address problems identified as a result of inadequate financial resources often frustrates health workers and managers, who are willing to bring about change; these are among the challenges that have been noted in decentralised settings in developing countries. Although local leaders have the power to make decisions that can improve service delivery, this decision-making space is limited by the resources that are available to them [38, 39]. The resources available to managers must therefore be expanded if they are to make significant changes towards improving service delivery.

Another factor that limited the ability of managers to make positive decisions was the power dynamics in the district. It has been noted that power dynamics can influence collaborative M&E practices [11]. Local political and technical leaders wield a lot of power in decentralised settings. Managers and leaders in other key positions are therefore often unwilling to make decisions that may spoil their relationships with such local leaders. A district health officer may therefore find himself unable to discipline a health worker who is closely related to a high-ranking district officer.

Other local contextual challenges included inadequacies in data collection and analysis, as well as report writing and information use at district offices and health facilities [40]. Some of the inadequacies were related to inadequate skills for checking the data collected and reported by the facilities to the districts, or to the inability to appropriately analyse the data collected. When we noted this, we planned to conduct a data quality assessment and to provide refresher training for the district biostatistician and records officers. Unfortunately, we were not able to address all these gaps because of financial and human resource constraints. Another related challenge was linked to the way key decisions were often made by managers and district leaders. Decisions about programming of public health programmes are often influenced not only by the data but also by the tacit knowledge of the programme implementers [41]. This therefore meant that there was low demand for data for decision-making both at facility and district level. Projects that aim to influence decision-making at district, community and facility level therefore need to budget funds for strengthening data collection, analysis and evidence generation. If district leaders have such training and an intrinsic desire to promote accountability, then they could spearhead similar activities that are geared at changing the status quo and improving service delivery.

The major strength of this paper is that it draws its data from several sources and therefore had adequate triangulation of data sources. Furthermore, the prospective nature of the M&E of activities also allowed the information obtained to be used in real time to improve the implementation of the study. The study also contributes to the needed literature on participatory M&E approaches, thereby demonstrating the value of stakeholder involvement in decision-making, and how and at what level to involve them. However, one limitation of the paper is that it does not indicate the actual amount spent on M&E activities despite them being very resource intensive. To promote sustainability of the approaches used herein, we suggest that similar programmes embed their data collection needs within existing routine systems of data collection so as to limit the additional cost of data collection. Similarly, feedback to stakeholders can be embedded within other existing stakeholder and programme meetings.

## Conclusions

Our implementation experience has revealed that a combination of participatory M&E approaches and feedback to stakeholders is very useful in tracking progress and identifying emerging implementation challenges, which help in facilitating planning and decision-making during implementation. Borrowing from our implementation experience, supporting districts to have crosscutting routine information and generating and sharing platforms that involve stakeholders from different sectors is crucial for the successful implementation of complex development interventions. However, there is a need to strengthen the skills of those responsible for the collection and analysis of data that is used to generate local evidence. Similarly, the resources required for addressing identified problems also need to be expanded so as to enlarge the decision-making space for key implementers and decision-makers.

Future research on participatory M&E could include documentation of resource needs, exploration of approaches to evaluate the effectiveness of stakeholder engagement and development of measures to assess the contribution of participatory M&E.

## Open peer review

Peer review reports for this article are available in Additional file 4.

## Additional files

**Additional file 1:** Changes in savings and transport for health. (DOC 31 kb)

**Additional file 2:** Data use for decision-making story of change. (DOC 30 kb)

**Additional file 3:** Changes management and leadership story of change. (DOC 26 kb)

**Additional file 4:** Open peer review reports. (PDF 430 kb)

## Acknowledgements
We are thankful to the district health officer and other stakeholders from the districts of Kamuli, Kibuku and Pallisa, who consistently attended the project review meetings and took appropriate decisions to address the maternal and newborn challenges in the three districts. We are also grateful to Future Health System and Makerere University School of Public Health for their technical support during the design and implementation of this study.

## Funding
This research and publication was funded by the UK Department for International Development through a grant (PO5467) to the Future Health Systems (FHS) Consortium. This study was funded in part by a grant from Comic Relief to the MANIFEST project.

## Availability of data and materials
Data sharing not applicable to this article as no datasets were generated or analysed during the current study.

## About this supplement
This article has been published as part of *Health Research Policy and Systems* Volume 15 Supplement 2, 2017: Engaging Stakeholders in Implementation Research: tools, approaches, and lessons learned from application. The full contents of the supplement are available online at https://health-policy-systems.biomedcentral.com/articles/supplements/volume-15-supplement-2.

## Authors' contributions
RMK carried out data collection, analysis and led the writing of the manuscript with the contribution from all authors. LP, EEK, GN and HNL participated in conceiving and reviewing the study. AB, GM and DN-B participated in reviewing the study results. SNK and AG participated in reviewing the study. DHP provided general guidance in study design, data analysis, and participated in drafting and reviewing the manuscript. All authors read and approved the final manuscript for publication.

## Ethics approval and consent to participate
Ethics approval for this paper was part of the approval provided to the MANIFEST study which was approved by the Makerere University School of Public Health Higher Degrees and Research Ethics Board and the Uganda National Council for Science and Technology.

## Consent for publication
Not applicable.

## Competing interests
The authors declare that they have no competing interests.

## 
## Author details
[1]Department of Health Policy Planning and Management, Makerere University School of Public Health, Kampala, Uganda. [2]Department of Social Policy, London School of Economic and Political Science, London, United Kingdom. [3]Department of International Health, Johns Hopkins University School of Public Health, Baltimore, MD, United States of America. [4]District Health Office, Kibuku District Local Government, Kampala, Uganda. [5]District Health Office, Pallisa District Health Office, Kampala, Uganda. [6]District Health Office, Kamuli District Health Office, Kampala, Uganda. [7]University of the Western Cape, Cape Town, South Africa.

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
