## [Open peer review reports. (PDF 430 kb) · Health Research Policy and Systems]

Title: Participatory Monitoring and Evaluation Approaches That Influence Decision-Making: Lessons from a Maternal And Newborn Study In Eastern Uganda

Reviewer 1: J. Bradley Cousins

Reviewer's report

General comments

The article is well written and provides an interesting account of a participatory approach to M&E in the context of the design and implementation of a complex health intervention in a development setting. The study is essentially an instance of research on evaluation (RoE); the authors describe in great detail the implementation of the participatory M&E activities and stand off the evaluation to reflect and comment on effects, facilitators, and challenges to the approach.

Regrettably, the authors do not really frame the report as RoE which leads to certain points of confusion: (i) the motivation for the study is implicit at best and (ii) the authors have a tendency to co-mingle data in service of the research with data gathered during the implementation of the M&E approach.

The article is well written and clear and ample evidence in the form of narrative with supporting tables and figures is provided to illustrate the observed M&E practice and alleged effects. But given that this is a research study the question guiding the research and the motivation for it are not well grounded in the extant literature on participatory and collaborative evaluation practice. Much has been learned and documented over time about justifications for engaging with collaborative forms of evaluation (e.g., pragmatic, political, philosophical), the purposes of such practice (practical, transformative) and the form participation takes in practice (e.g., control of evaluation decision-making, diversity in stakeholder selection for participation, depth of participation in the technical aspects of M&E). Principles for collaborative approaches to evaluation have been developed and validated. The present study does not converse well with this literature.

An elaborate explication of the methods for generating and capturing data in support of the research narrative is missing. Only vague references were made to two reflective meetings held, one with three contributors and a second with only two. From these reflections claims about effects of participation are made as the M&E process is described (many such claims appear in tables 3, 4 and 5). Yet in the absence of corroborating evidence it is impossible to judge the trustworthiness of the claims that are made. The data appearing in Box 1 is an example but nothing is said about how such data were captured. Finally, the paper does not conclude with hard hitting implications for (i) participatory M&E practice or (ii) ongoing research in the area.

Having said all of this, I am persuaded that the case is of high interest and potentially rich in insights that it can provide into doing and using participatory M&E. I liked the balance in the discussion between what worked well and what challenges emerged. I also liked the attention to unintended consequences of the intervention in the M&E process. I believe a serious revision of the paper would enable it to provide a significant contribution to the development evaluation literature.

Major compulsory revisions

I recommend that the authors attend to the following revisions in order to make the paper more publishable:

- Better frame the study as RoE and ensure adequate distinctions between data for research and evidence supporting the descriptive narrative of the actual M&E implementation
- Enrichen the literature review on participatory approaches to evaluation in order to better justify the question for research and to provide fodder for discussion and determination of implications for practice and research
- Provide added detail in the description of the M&E process to clarify its: justification (why was the participatory approach chosen in the first place, and by whom?); purpose (was the approach predominantly practical, [which seems likely]? To what extent were there transformative intentions [not much said about that]?); and form (who controlled decision making? Who selected stakeholders for participation? To what extent did the range of stakeholders participate in all aspects of the M&E process?)
- Work to provide support for the reflective claims that were made. For example, the article is co-authored by several people, presumably all involved at some level in the process and representing different stakeholder perspectives. Making clear the extent to which different stakeholder participants concur with the reflective narrative would help to support claims. Were there points of divergence of opinion? The data appearing in Box 1 also provide support for claims. Are there similar data available to be used in this way? Elaborate on how these data were collected.

Provide a thoughtful section on implications for M and E practice as well as an agenda for ongoing inquiry in this area. In doing so the contribution of the study will become much more apparent.

Minor essential revisions

No essential concerns. 'Data' should be plural.

Declaration of competing interests

None

Reviewer 2: Katharine Shelley

Reviewer's report

General comments

This paper succeeds at describing participatory M&E approaches useful in identification of implementation issues and subsequent problem solving through engaging a variety of stakeholders at the community- and district-levels. This research adds to a growing evidence-base around the importance of stakeholder engagement in data use for decision-making, and the participatory methods described are widely applicable to community- and district health systems strengthening initiatives.

Major compulsory revisions

1. More elaboration on capacity building is required given the thrust of the paper is on participatory M&E. For example, did the project team endeavour to strengthen the capacity of any local or district-level stakeholders to continue these participatory M&E approaches once the 3-year implementation period was over? If not, then more details should be provided about why capacity building was either outside the scope of the project and/or too difficult to incorporate. While "capacity building of leaders in management" is one of the key interventions of the MANIFEST project (p.5), the authors have only made brief mention of capacity building (p.17). The paper would be greatly improved by tackling this topic in more detail.

Data triangulation is briefly mentioned, and supported by evidence, at the beginning of the discussion. Subsequently, the authors suggest triangulation is THE key strength of the paper. If it is the key strength, then I suggest introducing triangulation earlier in the paper, in methods and/or results, to set the reader up for how the various M&E approaches come together to triangulate findings and support decision making. For example, at the quarterly stakeholder meetings were there explicit efforts to triangulate data from multiple sources? As currently written, the M&E approaches appear largely standalone pieces of the larger project.

Minor essential revisions

As indicated in the comments embedded within the Word document, I have noted where minor revisions would help to clarify content and improve the overall flow of the manuscript. Because there are numerous minor comments, I have not listed them individually here.

Declaration of competing interests

I declare I have no competing interests.

Please note I am a recent PhD graduate of the International Health Department at Johns Hopkins School of Public Health, to which several of the co-authors are affiliated. I have collaborated with Ligia Paina, Asha George, and David Peters on several projects, but I am not connected in any way to the research in Uganda upon which this paper is based nor to the work of *Future Health Systems*.

**Reviewer 2:** Katharine Shelley

**Reviewer's report – comments on manuscript**

**Participatory Monitoring and Evaluation Approaches That Influence Decision-Making:**
**Lessons from a Maternal And Newborn Study In Eastern Uganda**

**Authors:** Rornald Muhumuza Kananura^{1*}, Elizabeth Ekirapa Kiracho¹, Ligia Paina², Ahmed
Bumba³, Godfrey Mulekwa⁴, Dinah Nakiganda-Busiku⁵, Htet Nay Lin Oo², Suzanne N.
Kiwanuka¹, Asha George^{2,6}, David H. Peters²

* Corresponding Author

1. Makerere University School of Public Health, Department of Health Policy Planning and
Management, Kampala Uganda

2. Johns Hopkins University School of Public Health, Department of International Health,
Baltimore, MD

3. District Health Office, Kibuku District Local Government

4. District Health Office, Pallisa District Health Office

5. District Health Office, Kamuli District Health Office

6. University of Western Cape, South Africa

Rornald Muhumuza Kananura: mk.rornald@musph.ac.ug

Elizabeth Ekirapa Kiracho: Ekky@musph.ac.ug

Ligia Paina: lpaina@jhu.edu

Ahmed Bhumba: drbumba2012@gamil.com

Godfrey Mulekwa: gmulekwa@gmail.com

Dinah Nakiganda- Busiku: dnbusiku@hotmail.com

Htet Nay Lin Oo: hoo1@jhu.edu

Suzanne N Kiwanuka: skiwanka@musph.ac.ug

Asha George: asgeorge@uwc.ac.za

David H Peters: Dpeters@jhu.edu

1 **Abstract**

[revised manuscript text omitted]

Comment [SK1]: Here you've jumped from household level to facility level. It might be worthwhile to include a sentence about the "community" level given the focus on VHTs and community providers in this paper.

mother and her newborn [8, 9]. Addressing these barriers to access should be informed by
periodic collection of data that tracks implementation changes and challenges, which can be
shared regularly/systematically with community stakeholders (such as community health
workers (village health teams) and community local leaders), health service providers, and
decision makers at district and national level. Translating implementation findings for
stakeholders enables them to gain a better understanding of the intervention and its possible
effects [10, 11]. This helps in engaging stakeholders in defining the problem and availing the
solutions to address identified problems [2, 9, 11, 12]. Furthermore, information sharing with
local stakeholders helps to redesign and improve programs that do not reach their intended
beneficiaries [13, 14]. This strategy also regularly connects decision makers, implementers and
researchers and promotes accountability to their constituent communities. For example, it is
important to be able to track whether health workers are adhering to national and international
guidelines and if they are not to discuss and agree what can be done to mitigate it.

We used a range of participatory monitoring and evaluation approaches during the
implementation of a maternal and newborn health project called Maternal and Neonatal
Implementation for Equitable Systems (MANIFEST) in three districts of Eastern Uganda from
2013 to 2015. To encourage flexibility in how the intervention could be implemented over time,
and to be able to respond to the changing concerns of stakeholders, we opted to use M&E
methodologies that collect information beyond the key outcomes and process/input indicators,
such as unanticipated project implementation changes and challenges, while also paying
attention to understanding stakeholders and their influence.

**The MANIFEST project**

Comment [SK2]: For readers unfamiliar with Uganda's health system, it would be helpful to define the VHT. Is it synonymous with CHWs?

MANIFEST had several key interventions, which were implemented using a participatory action
research approach. They included 1) community mobilization and empowerment through the
community health workers home visits, community dialogue meetings, radio talk shows and
messages; 2) improvement of financial and geographical access to care by promoting savings for
delivery care and organizing local transport; and 3) health systems strengthening through training
of health workers, mentorship, supportive supervision and capacity building of leaders in
management. These interventions were provided only in the intervention area except for the
radio talk shows and messages, which were aired on radios with listenership in the control areas
as well and support supervision, which was routinely provided by the district health team in both
the control and intervention area. More details about the intervention are available in the
MANIFEST study design paper [15]

The MANIFEST project had a multisectoral group of stakeholders who played different roles.
The research team comprised of members from the district level (district health officers, and
district reproductive health focal persons) and researchers from the Makerere University School
of Public Health and Johns Hopkins University School of Public Health. They were responsible
for designing the intervention and implementing it in collaboration with the other district, Sub
County and community level stakeholders.

The community level stakeholders included men and women of reproductive age, local transport
providers, saving group leaders and village health team (VHT) members. The men and women
of the community were important stakeholders; since they made decisions about seeking
appropriate care for mothers and newborns and preparing for birth by ensuring that they had the
financial resources required in addition to planning transport and purchasing other requirements

Comment [SK3]: Suggest re-ordering this community stakeholder list according to the order you describe the stakeholders: men & women; VHT members; savings group leaders; and transporters.

needed for the mother and newborn. The VHT members were responsible for doing home visits
and conducting community dialogues, which were community meetings established to discuss
maternal and newborn health issues. Saving group leaders and transporters provided relevant
services that contributed to increasing access to cash and transport for maternal and newborn
health. The sub county and district level stakeholders comprised of the health workers, various
community leaders and decision makers (religious leaders, political leaders and technocrats). The
health workers, facility and district managers were responsible for ensuring that quality services
were provided while the **policy makers** at sub county and district level were responsible for
providing oversight and ensuring that required decisions were made about maternal and newborn
health during sub county and district council meetings or other such fora.

Comment [SK4]: Is this equivalent to “decision makers” in prior sentence?

During study implementation, the research findings were analyzed, synthesized, and shared
**regularly** with the different stakeholders in the intervention area. The purpose of this paper is to
examine how the participatory M&E approaches used were able to identify emerging
implementation **issues**, and how they influenced decision making by community and district
level stakeholders.

Comment [SK5]: It could help to define how regularly, on average, this occurred... i.e. quarterly? Bi-annually? As needed when key findings emerged?

Was this operating as continuous learning platform type of approach?

**Methods**

***Study area and design***

The MANIFEST study was conducted in three rural districts of Kamuli, Kibuku and Pallisa in
Eastern Uganda. The estimated population in this area is 1,106,100 (Kamuli-500, 200, Kibuku-
209, 000 and Pallisa-396, 900) [16]. The three districts have 104 health facilities, 33 in Pallisa,
17 in Kibuku and 54 in Kamuli [16]. **In these areas**, only about 1 of 2 **pregnant** women attend

Comment [SK6]: By “issues” here do you mean the M&E approaches focused on emerging implementation challenges and bottlenecks? Or did the M&E also explore best practices as “issues”? Maybe this is clarified below, I’ll keep reading!

Circling back to this point – in Table 1 there appears to be a focus on “lessons learned” at the quarterly meetings, but that wasn’t obvious to me until getting to the table.

Comment [SK7]: Is this the DHS figure for the districts? Region?

1 four or more antenatal care visits or deliver in health facilities, which is less than the nationwide
2 average [17]. The MANIFEST baseline study estimated the neonatal death rate to be 34 per
3 1,000 live births [18].

Comment [SK8]: Please state the nationwide average so the reader has a comparison.

Comment [SK9]: Mortality?

Comment [SK10]: Suggest also stating the DHS estimated NMR for this area of the country, or at least at the national level.

***M&E approaches and stakeholder involvement***

The data for this paper is drawn from retrospective reflection on the various M&E approaches
described in the forthcoming section. Two reflection meetings were held to discuss the benefits
of using the participatory ~~M&-and-E~~ approaches and to discuss how they influenced decision
making. The first meeting was attended by three of the authors of this paper, while the second
was attended by only two of the authors. Figure 1 provides a summary of the ~~M&-and-E~~ data
collection approaches and tools that were employed as well as the stakeholder engagements that
were undertaken.

Comment [SK11]: Were the reflection meetings just with the three authors (1st meeting) and the two authors (2nd meeting)... or conducted with stakeholders?

If the former, I am wondering if it warrants any mention at all? I would consider meetings between the researchers only as part of the writing process (essentially outlining the main benefits of using participatory M&E and identifying how such efforts resulted in improved decision making. However, if stakeholders were engaged and present during these retrospective reflections then very much worth noting.

Comment [SK12]: Change to M&E for consistency with how you've referred to it throughout most of the paper.

Comment [SK13]: Figure 1 is a useful visual. Please see suggestions attached to the Figure on how to make it stronger.

**Figure 1**

*Stakeholder involvement*

At the planning and the design stage, a planning meeting that involved the research team
members, health providers, district leaders, sub county leaders and community members was
conducted in order to identify community conditions/problems that lead to underutilization of
maternal health services and contribute to maternal and newborn deaths. During the planning
meeting, the stakeholders were asked to discuss how to address the problems identified using
available resources and a given time frame. The involvement of the stakeholders at the planning
stage provided a better understanding of the maternal and newborn problems and guided the
selection of interventions that were implemented.

During the implementation phase, the stakeholders at the community and sub county levels in the
intervention areas were engaged in addition to the district level stakeholders. They were engaged
through quarterly group meetings that happened at sub county and district level, quarterly
support supervision visits to the health facilities and quarterly group meetings with the VHTS
and the communities (community dialogues). These stakeholder meetings have been described in
**T**able 1. Results from study household surveys, health facility support supervision reports, key
informant interviews, focus group discussions with the stakeholders/beneficiaries were presented
at the stakeholder meetings. Based on the presentations and discussions, appropriate actions were
then taken by district planning leaders, health workers, health managers and the research team.

Table 1

*Quantitative data collection*

Quantitative information was collected through household surveys, health facility support
supervision visits, health information utilization data and reports from the community health
workers. We conducted household surveys at baseline, mid-term, and end line so as to determine
changes in the study outcomes, while we used Lot Quality Assurance Sampling (LQAS)
techniques to conduct quarterly household surveys during the first nine months of the study to
monitor the uptake of key intervention elements. The main outcomes for LQAS household
surveys were changes in facility deliveries, ANC attendances, birth preparedness practices, and
knowledge of birth preparedness, pregnancy, labor and newborn danger signs. Every quarter, we
randomly selected 5 villages as supervision areas in each district (supervision units), from which
we randomly sampled 19 eligible households for assessment. A team of 5 district-based persons

Comment [SK14]: Staff? Unclear are the five all district/government employees? Or study staff. Suggest rephrasing for clarity.

who included the biostatistician and HMIS focal person collected the data. Table 2 provides
details of these data collection methods.

Table 2

*Qualitative data collection*

Comment [SK15]: Who collected the qualitative data via FGDs and KIIs?

The qualitative data were collected through focus group discussions and key informant
interviews and quarterly review meetings at district and sub county level. They are described in
more detail in table 2.

*Mapping and Theory of change*

Comment [SK16]: I suggest moving this section up to right after the "stakeholder involvement" sub-paragraph. It appears to flow more naturally from there.

At the design stage, stakeholders were consulted so as to identify local problems and feasible
local innovations to address the identified problems. This stage guided the team to map out the
possible study outcomes, influential stakeholders to be targeted, partnerships to be identified,
strategies for addressing community and health providers' behaviors, and inputs needed for the
implementation of different strategies. This information was used to develop a theory of change.
The theory of change enabled the research team members to clarify not only the ultimate
outcomes and impacts they hoped to achieve but also the avenues through which they expected
to achieve them. This helped the research team and the local stakeholders build consensus on the
implementation pathways. More details about the theory of change and how it was used are
available in Paina et al (19).

Comment [SK17]: Repetitive with the stakeholder involvement section. If you move this section earlier to appear after "stakeholder involvement", this sentence can be deleted.

*Most significant change*

Comment [SK18]: I suspect the ToC is presented in this paper, but it might be helpful to also include it here... even if just a simplified version.

We used a modified version of the most significant change approach (MSC) to help us track the
most significant changes experienced by the health providers and the community during the
implementation phase[20] (Fig 1). We did this by collecting stories of change during focus
group discussions with the community, key informant interviews with health providers and local
leaders, and meetings (quarterly meetings, health workers symposia, and research team
meetings). The stories spanned across several domains that included quality of care provided at
the health facilities, health workers' attitudes, changes in health care management/leadership
skills and behavioral changes among mothers in terms of birth preparedness and newborn care.
We however did not rank these stories so as to identify the most significant change; rather we
considered all of them as stories of change since our aim was to capture perceptions of change
from the stakeholders view point.

*Participatory impact pathway analysis (PIPA)*

We used participatory impact pathway analysis (PIPA) to identify key stakeholders involved in
maternal and newborn health. The PIPA workshop was conducted in the first and second year of
implementation. Details about how it was conducted are available in Ekirapa Kiracho et al (21).
We used PIPA to analyze the type, role, and strength of each stakeholder, as well as how they
were connected with one another in the context of maternal and newborn services. This helped
the project team to understand the actors in maternal and newborn health, the resources that they
possessed as well as the power and influence that they had in promoting achievement of the
project objectives.

**Results**

In the subsequent sections, we present findings that illustrate how M&E information shared with
each group of key stakeholders was linked to the decisions or actions taken.

***Community level***

During the design phase of the program we held focus group discussions and stakeholder
meetings with local stakeholders who included women, men, transporters, saving group leaders,
district leaders and health workers. The purpose of these discussions were to identify local
problems and feasible solutions, existing local resources including existing structures, human as
well as financial resources. Through the discussions we were able to identify the problems that
affect maternal and newborn health services in three main areas. The areas included birth
preparedness; transport; and quality of MNH care services in the health facilities. The problems
related to birth preparedness included: lack of awareness of its importance, negative cultural
practices, men neglecting their roles, lack of knowledge about family planning, poor saving
culture and poverty. The transport problems included: absence of ambulances, long distances to
health units, lack of appropriate transport vehicles and high transport fares. The quality of care
was being compromised by frequent essential drug shortages, inadequate number of delivery
beds, understaffing, poor health workers' attitudes, irregular support supervision, staff
absenteeism, informal charges and poor technical and managerial skills. This information was
used to identify the interventions that were implemented. For instance, to address the challenge
of low awareness about the importance of birth preparedness, home visits by community health
workers were suggested and later included as one of the key interventions. To address poor

Comment [SK19]: Suggest deleting as you've already described who the community stakeholders include.

Comment [SK20]: Please clarify – do you mean governance structures?

managerial and technical skills, refresher training for health workers was proposed and provided
as one of the interventions for health system strengthening.

The, local resources that we identified included existing structures such as the sub county
committee, community development officers, local transporters, savings groups, radio stations
and VHTs. These resources were subsequently deployed during the project implementation,
which employed a participatory approach.

Comment [SK21]: Not clear. An example might help clarify.

During the implementation phase, we shared information about uptake of the intervention
elements and progress with implementation of the intervention with the community level
stakeholders. Table 3 provides a summary of key issues that were identified at the community
level and shared with community stakeholders, as well as the actions that were recommended by
these stakeholders.

Table 3

Data from the household surveys provided information about the uptake of various aspects of the
intervention. For example, in some of the hard-to reach areas, newborn deaths were high and
most of the women were delivering at home with assistance from traditional birth attendants.

Data collected from community health workers also helped the research team and district health
office capture the number of newborn deaths and maternal deaths more completely and
accurately. Previously the district only had data from the facility, which reflected a much smaller
number of maternal and newborn deaths. The focus groups were used to explore the reasons
behind these home deliveries and newborn/maternal deaths in more depth and to identify
possible solutions that could be undertaken by community, facility or district level stakeholders.

Table 4 provides a summary of the ~~main factors contributing to maternal and newborn main~~
~~circumstances surrounding the~~ deaths and solutions that were proposed.

Table 4

The main factors ~~causing maternal and neonatal deaths~~ included delays in deciding to seek care,
inadequate care at the health facilities with delays in deciding to refer mothers at the health
facilities. Some of the problems that had been identified during the problem identification phase
were still present even at the design ~~phasing~~ of the study. Their persistence during the
intervention showed that more attention needed to be given to addressing them. These issues
were then brought to the attention of Local leaders, health providers including VHTs, and district
planners in the community. For example, through the community dialogues, we emphasized the
importance of delivering in health facilities and preparing for birth by saving money so that
transport could be availed in case a mother was referred to a more specialized facility. We also
emphasized the importance of monitoring mothers using a partograph so that delays in labour are
detected early and referrals done on time.

As alluded to earlier we did surveys with the VHTs; to identify their knowledge about danger
signs and areas of weakness in conducting health education and referral during home visits.
Results from the second monitoring data collection exercise (6 months after the intervention
started) during which interviews were done with VHT's, revealed that only 46% knew at least
three newborn danger signs, ~~signifying low level of knowledge about newborn danger signs.~~
Furthermore only 29% of the VHTs ~~were well versed~~ ~~with the transport and savings component~~
of the intervention (Table 3). These results were shared with the VHT's during the quarterly
group meetings and refresher training was provided in these weak areas. The VHT's also ~~did~~

Comment [SK22]: Suggest removing. Implied.

Comment [SK23]: Unclear what you mean by "well versed" here.

[revised manuscript text omitted]

Comment [SK24]: This is the first you touch on capacity building, which is great, but also makes me wonder if there were capacity building elements to the other participatory M&E approaches described?

Comment [SK25]: Very helpful illustrative example

Discussion

This paper describes the participatory M&E methodologies and tools used to identify key
implementation issues and solve problems and how they influenced decision making. Use of a
combination of M&E approaches and tools had several benefits. Firstly, they allowed
triangulation of data from different sources leading to more complete reporting and a better
understanding of some of the issues noted. This allowed the stakeholders to get a comprehensive
picture of how different factors were interacting to influence maternal and newborn health
outcomes. For example several newborn deaths were reported in the intervention area. Hence, it
was important for the district health management team, health workers and other key

stakeholders to understand the circumstances that led to the deaths, so that measures could be put
in place to stop similar occurrences. The qualitative interviews that were done with women who
had lost their babies therefore aided in the identification of the factors that contributed to these
deaths. The district and sub county level stakeholders were then able to take actions to solve
some of the problems identified. Other researchers have also indicated the importance of
combining quantitative assessments of pre-specified mediating variables with qualitative
investigation of participant responses in testing and redefining the causal intervention
assumptions [9, 13, 22, 23].

Secondly, combining qualitative and quantitative data collection methods also allowed
identification of unanticipated pathways, and in-depth exploration of pathways which are too
complex to be captured using one method [22]. For instance to encourage male involvement,
facilities prioritize women who come with their partners and sometimes decline to work on
women who do not come with their partners during antenatal care. However we noted that this
becomes a barrier to seeking formal delivery care services for women who have no partners.

These women feel discriminated against and decide to shun all the facility services, as described.
~~This was the reason given for home delivery~~ by a woman who gave birth at home and later lost
her baby.

Comment [SK26]: Forgo?

Comment [SK27]: Could you end this paragraph by explaining the solutions/decisions that resulted from these findings from triangulating multiple sources?

Thirdly the frequent interactive monitoring of the implementation allowed us to identify gaps in
implementation and to identify practical solutions that could be implemented by those who were
responsible for improving service delivery. During supportive supervision, we realized that
whereas some of the implementation was constrained by factors that were beyond the control of

| the facilities or health workers, such as inadequate essential equipment and supplies and skilled
| motivated health workers [24–26], in several cases there was something that the health workers
| could do. For example some newborn babies were dying without being resuscitated and
| partographs were not being filled in some facilities simply because the health workers lacked the
| skills to do so. Availing information on these health facility gaps, and emphasizing the fact that
| the health workers could change this situation if supported, encouraged the facility and district
| management to take action whenever there was a problem that they could solve. Other
| researchers have indicated the importance of engaging clinical and management staff in
| discussion of implementation barriers and facilitators [25–27]. However for this monitoring to
| lead to improvement in service delivery, the gaps must be clearly specified, actions that are to be
| taken to mitigate them must be identified and persons responsible indicated, and follow up must
| be done to ensure that the required action was taken, otherwise the problems simply continue to
| persist.

Comment [SK28]: Very well articulated.

| Lastly, the participatory M&E methods ~~that we used~~ promoted interaction and dialogue between
| the stakeholders. We realized that the dialogue enhanced the ability of stakeholders to hold each
| other accountable, which was an unanticipated positive outcome. Other studies have also
| emphasized the need for information sharing with stakeholders at each stage of implementation
| [5, 9], which strengthens appropriate decision making, advocacy and resource sharing [12, 28,
| 29]. We however noted that in the absence of the research team, things tended to slide back to
| business as usual implying that strengthening of such accountability processes requires time and
| local champions before it can become entrenched into local systems.

Comment [SK29]: Jargon. Suggest rephrasing. What is meant by “business as usual”... that dialogue lessened? And/or then the accountability reduced?

In spite of the above benefits, we note that there were several challenges that may hinder use of
participatory M&E approaches for decision-making especially in low and middle-income
countries. One of the major challenges was that often the available resources were not adequate
for taking the required actions. One of the key weaknesses noted was in the referral system,
which needed a comprehensive set of communication and transport facilities at the community
and the facility level. Although some progress was made through the purchase of a motorcycle
ambulance and trailers, and purchase of motorized ambulances by politicians these were not
enough. Persistent inability to address problems identified as a result of inadequate financial
resources often frustrate health workers and managers who are willing to bring about change.
These are one of the challenges that have been noted in decentralized settings in developing
countries. Although local leaders have the power to make decisions that can improve service
delivery, this decision making space is limited by the resources that are available to them (30).
The resources available to managers must therefore be expanded if they are to make significant
changes towards improving service delivery.

Another factor that limited the ability of managers to make positive decisions was the power
dynamics in the district. Local political and technical leaders wield a lot of power in
decentralized settings. Managers and leaders in other key positions are therefore often unwilling
to take decisions that may spoil their relationships with such local leaders. A district health
officer may therefore find himself unable to discipline a health worker who is closely related to a
high-ranking district officer.

Other challenges included inadequacies in data collection and analysis, report writing and
information use at district offices as well as health facilities [31]. Some of the inadequacies were
related to inadequate skills for checking the data collected and reported by the facilities to the

districts. Another included inability to analyse the data collected appropriately. When we noted
this we planned to conduct a data quality assessment and to provide refresher training for the
district biostatistician and records officers. Unfortunately we were not able to do this because we
had only one monitoring and evaluation officer, and did not have adequate funds for conducting
the quality assessment, training the district biostatistician and providing continuous supportive
supervision of their work. Another related challenge was linked to the way key decisions were
often made by managers and district leaders. These decisions were often not driven by the data
but rather by the tacit knowledge of the stakeholders (ref- 32). This therefore meant that there
was low demand for data for decision making both at facility and district level. Projects that aim
at influencing decision making at district, community and facility level therefore need to budget
funds for strengthening data collection, analysis and evidence generation. If district leaders have
such training and an intrinsic desire to promote accountability then they could spear head similar
activities that are geared at changing the status quo and improving service delivery.

Comment [SK30]: Perhaps this level of detail is not necessary. I wonder if this could be summarized under the challenge of capacity building around data use for decision making and a lack of resources within the project to adequately carry out the required capacity building needs.

Comment [SK31]: Not listed in reference lists – but I’m curious to see the citation and it looks like an interesting paper!

The major strength of this paper is that it draws its data from several data sources and so there
was adequate triangulation of data sources. However, one limitation of the paper was that it does
not indicate the actual amount spent on M&E activities and yet these activities were very
resource intensive. To promote sustainability of the approaches used in this paper we suggest
that similar programs embed their data collection needs within existing routine systems of data
collection so as to limit the additional cost of data collection. Similarly feedback to stakeholders
can be embedded within other existing stakeholder and programme meetings.

Comment [SK32]: The paper doesn’t really focus on triangulation of sources – in fact, it’s only mentioned once in the beginning of the discussion.

What about other strengths? You seem to dwell mostly on weaknesses... I like the sustainability angle and suggest this could be expanded.

What about the prospective nature of the participatory M&E as a strength of the approach?

21

22

1 **Conclusions**

[revised manuscript text omitted]

45

- 16. Uganda Bureau of Statistics. 2014 Statistical Abstract. Kampala; 2014.
- 17. Uganda Bureau of Statistics. Uganda demographic and health survey 2011. Kampala; 2011.
- [http://ubos.org/onlinefiles/uploads/ubos/pdf_documents/Uganda DHS 1988-89 Final Report.pdf](http://ubos.org/onlinefiles/uploads/ubos/pdf_documents/Uganda_DHS_1988-89_Final_Report.pdf).
- 18. Kananura RM, Tetui M, Mutebi A, Bua JN, Waiswa P, Kiwanuka SN, et al. The neonatal
- mortality and its determinants in rural communities of Eastern Uganda. *Reprod Health*. 2016;:1–
- 9. doi:10.1186/s12978-016-0119-y.
- 19. Paina L, et al " Using Theories of Change to inform implementation of health systems
- research and innovation: experiences of Future Health Systems consortium partners in
- Bangladesh, India, and Uganda (submitted to the same supplement)
- 20. Davies R, Dart J. The “Most Significant Change” (MSC) Technique. 2005.
- 21. Ekirapa -Kiracho E, Upasona G, Ritika B, Paina L Engaging stakeholders: Lessons from the use of
- participatory tools for improving Maternal and Child Care Health Services.
- 22. Moore GF, Audrey S, Barker M, Bonell C, Hardeman W, Moore L, et al. Process evaluation
- of complex interventions: Medical Research Council guidance. *BMJ*. 2015;350 mar19 6:h1258–
- h1258. doi:10.1136/bmj.h1258.
- 23. de Vlaming R, Haveman-Nies A, Van't Veer P, de Groot LC. Evaluation design for a
- complex intervention program targeting loneliness in non-institutionalized elderly Dutch people.
- *BMC Public Health*. 2010;10:552. doi:10.1186/1471-2458-10-552.
- 24. Chaillet N, Dubé E, Dugas M, Francoeur D, Dubé J, Gagnon S, et al. Identifying barriers and
- facilitators towards implementing guidelines to reduce caesarean section rates in Quebec. *Bull*
- *World Health Organ*. 2007;85:791–7. doi:10.2471/BLT.06.039289.
- 25. Evans-Lacko S, Jarrett M, McCrone P, Thornicroft G. Facilitators and barriers to
- implementing clinical care pathways. *BMC Health Serv Res*. 2010;10:182. doi:10.1186/1472-
- 6963-10-182.
- 26. Camden C, Swaine B, Tétreault S, Carrière M. Going beyond the identification of change
- facilitators to effectively implement a new model of services: lessons learned from a case
- example in paediatric rehabilitation. *Dev Neurorehabil*. 2011;14:247–60.
- doi:10.3109/17518423.2011.577049.
- 27. Addington D, Kyle T, Soni D, Wang J. Facilitators and barriers to implementing quality
- measurement in primary mental health care: systematic review. *Can Fam Physician*.
- 2010;56:1322–31. <https://www.ncbi.nlm.nih.gov/pmc/articles/PMC3001932/>.
- 28. Schultz JA, Pandya S, Sims M, Jones JA, Fischer S. Participatory monitoring and evaluation
- within a statewide support system to prevent adolescent substance abuse. *J Prev Interv*
- *Community*. 2013;41:188–200. doi:10.1080/10852352.2013.788347.
- 29. Braithwaite RL, McKenzie RD, Pruitt V, B.Holden K, Katrina Aaron M, Chavone Hollimon
- 36 M. Community-Base Participatory Evaluation: The Health Start Approach. *Health Promot Pract*.
- 2013;10:54–6. doi:10.1177/1524839912443241.
- 30. Bashaasha B, Mangheni NN, Nkonya E. Decentralization and rural service delivery in
- Uganda. IFPRI Discussion paper 01063. International Food Policy Research Institute. Accessed
- on
- 10th.06.2017[http://ebrary.ifpri.org/utils/getfile/collection/p15738coll2/id/124890/filename/124891](http://ebrary.ifpri.org/utils/getfile/collection/p15738coll2/id/124890/filename/124891.pdf)
- [.pdf](http://ebrary.ifpri.org/utils/getfile/collection/p15738coll2/id/124890/filename/124891.pdf)
- 31. Hotchkiss D, Diana M, Foreit K. How Can Routine Health Information System Improve
- Health System Function in Low-Resource Setting. Chapel Hill; 2012.
- <https://www.measureevaluation.org/resources/publications/sr-11-65>.

1 **Table 1: Description of stakeholder involvement**

[revised manuscript text omitted]

Key issue identified	Solutions proposed
Delay in deciding to seek care for ANC and delivery Delay in deciding to refer the mother to hospital "The first time she attended ANC, she was advised to go to the hospital. However, she never went because she thought using the local herbs would cure her. When the time for delivery reached, she went to HC III (immediately the labor started-8:00am). When the facility staff failed, they referred her to the Hospital at 11:00Pm (at night). The hospital opted for a caesarian. After the operation,	Religious leaders, community health workers, and local leaders to continue participating in sensitizing their communities on the importance of accessing maternal health services from health facilities during home visits and community dialogues Strengthen monitoring of women in labour using partographs through mentorship and support supervision so that referrals are not delayed

Comment [SK33]: Is there any tracking to know whether these solutions were enacted?

Key issue identified		Solutions proposed
she bled too much and this resulted in her death. Fortunately, the baby survived". Deceased's Sister		
Poor health worker attitudes "I went to the facility when my pregnancy was 2 months but was denied access to services because I had not gone with my husband. I again went there when it was 6 months and the same happened. I tried to explain to the health worker but she could not listen to me. When the time for delivery reached, I decided to deliver from home because I feared to go back to the facility. Two days after delivery, my child died". Mother 35 Yrs., gravid 5 and above		Health workers advised to relax the policy of only working on women who attended antenatal care with their partners
Delay in deciding to refer the mother Lack of immediate transport for referral "I reached the HC at 2:00PM but was referred to Hospital at 3:00am... the health workers found that they could not manage me and I was referred to the regional referral Hospital. Unfortunately, the driver for the ambulance was not around, The vehicle was got at 4:00am.... When I reached Regional Hospital a decision was made to do a caesarian. Unfortunately the baby died immediately after delivery". Mother, 29 Yrs., gravid 5 and above		Strengthen monitoring of women in labor using partographs through mentorship and support supervision so that referrals are not delayed District health office to work with CAO to make sure the ambulance driver and fuel are always available to ease referral.
Lack of health worker skills in managing obstructed labor "I attended ANC four times at HC III. During delivery, the baby's head came out but other parts could not come out. I tried to push but it could not come out. Unfortunately, it died before even coming out. I think it was too big". Mother 24 yrs., gravid 3		Obstetricians and gynecologists to continue mentoring midwives on how to handle complications during delivery through mentorship

1

2 **Table 5: Health facility level information and actions taken**

Emerging issues	Data collection methods and avenues for information sharing	Actions suggested and taken
Monitoring of women in labour		
Limited use of partographs to monitor the progress of labor	Information collected through supportive supervision visits and shared through district review meetings	Training of the health workers on the use of the partograph through mentorship program and support supervision. Training was done and health facilities started putting aside money for buying partographs.
Maternal and newborn death high in some health facilities	Data was collected through records review and shared during quarterly review meetings	Maternal and newborn death audits were recommended. The District reproductive health focal person found that in one hospital the nurses did not know how to resuscitate newborns, so she did refresher training. In another facility unnecessary augmentation of labor was being done leading to foetal distress and stillbirths so the midwife was given guidance about when to augment labor.
Care for newborns		
Poor care of small babies – neonatal resuscitation and using Kangaroo Mother Care	Data was collected through household surveys and shared during quarterly review meetings	Health workers were trained on how to care for small babies through the mentorship program. Newborn care corners started at the health facilities. Pediatrician was added to the mentorship team so as to improve newborn care.
Resources for providing maternal and newborn services		
Stock out of maternal and newborn essential drugs and supplies	Information collected through supportive supervision visits and shared through	Training the health facility managers on proper drug requisitioning during the certificate course on management. Facilities that had excess shared with

	district review meetings	facilities that had inadequate amounts.
Four health facilities did not have a placenta Pit for disposal of placentas.		The sub county leadership was informed at the sub county review meeting and they availed funds to construct the placenta pits
Some hospitals and health center IVs did not have an ambulance	Where is the content for this box?	Political leaders to lobby politicians and other stakeholders to buy ambulances- members of parliament in Pallisa district bought 4 motorized ambulances. One sub county bought a motorcycle ambulance Fundraising was done and 10 trailers for motorcycle ambulances purchased
Ambulances have mechanical problems and cannot transport women		Medical superintendent for the hospital was asked to ensure funds allocated for repair of the ambulance during district review meeting and this was done (Pallisa district).
No fuel for the hospital ambulance		The district health officers availed money for fuel for the ambulance from his budget line at district level (Kibuku district)

Comment [SK34]: Is this to be blank?

| Figure 1: M&E tools, approaches, and activities used at different stages of program design and implementation.

Comment [SK35]: This is a very helpful visual!

I have a couple of comments and suggestions on improving the visual.

1. Is the "Stakeholder categories" box across the top meant to include all types of stakeholders were engaged throughout the course of the design and implementation phases? Should all stakeholder categories engaged be listed in each type of M&E approach?

2. It looks like MSC was repeated annually, or was an ongoing activity throughout. Therefore, is it necessary to repeat that the text throughout the diagram? Should the data collection tools and stakeholder engagement activities? Should we eliminate the repetitive text by using asterisks or other indicators to denote that it was repeated (similar to PIPA)?

3. The dotted arrows leading from the PIPA line are distracting, are they necessary to the diagram? We know that PIPA occurred in Year 2, probably sufficient detail to know it occurred in Years 2 and 3.

4. What is the purpose of the quarterly brackets in the implementation area of the graphic? – just to emphasize quarterly meetings occurred?

5. Where is the "quantitative data collection" in this graphic? – please describe the quantitative data collection.

1 **Box 1: Data use for decision making Story of Change**

2

“.....as a result of MANIFEST study, we now use a lot of our data in planning and budgeting. For example, not long ago we did not have adequate resources to construct maternity wards in every Sub-county, so we had to use our data and we said ok, which place has a biggest ANC deliveries, which place is having big out-patient attendances. We then decided that we have the general ward constructed in Kadama health center III, in Kadama Sub-county. So, we are now using our data because it is now available contrary to what was there before, where you would ask like how many delivery do you have on average per month and you’re like a aaa---- oba this number [guessing]. But now we can easily check, all the health indicators because we have data center where all our information is readily available. So, we can use our data for planning and decision making, and even staff allocation. For example we decided to allocate more midwives and other health workers in facilities that have high number of ANCs/deliveries and outpatient respectively. In addition, we have use this information to justify the need for health works, which has convinced the Ministry of health to consider relaxing the ban on the recruitment of health workers.”

12

13

14

15

Author's response to reviews

We have responded to the reviewers' comments and we hope the paper has now improved

Reviewer 1: J. Bradley Cousins

1. Better frame the study as RoE and ensure adequate distinctions between data for research and evidence supporting the descriptive narrative of the actual M&E implementation

Thank you very much for this comment. We have revised the manuscript and tried to frame it more as research on evaluation.

2. Enrichen the literature review on participatory approaches to evaluation in order to better justify the question for research and to provide fodder for discussion and determination of implications for practice and research

Thank you very much for this advice. We have provided more literature on participatory M&E approaches see lines 12- 23 (pg 4), lines 1-20 page 5 and Line 1-10 page 6

3. Provide added detail in the description of the M&E process to clarify its: justification (why was the participatory approach chosen in the first place, and by whom?); purpose (was the approach predominantly practical, [which seems likely]? To what extent were there transformative intentions [not much said about that]?); and form (who controlled decision making? Who selected stakeholders for participation? To what extent did the range of stakeholders participate in all aspects of the M&E process?)

Thank you very much for this advice. We have revised the description of the M&E process by pointing out the reasons for the participatory approach (Page 10/11 under M&E Approaches' section). The participatory approach was mainly practical. Although the project aimed at addressing the needs of the marginalised, we feel that the transformative component was mainly in relation to empowering the community with knowledge about maternal and newborn health and with information and the means to improve their financial preparedness. We have also indicated how participatory the approach was by indicating the roles of various stakeholders (page 10/11 under stakeholders involvement section) and how different stakeholders were engaged at different levels (page 10/11 and Figure 1).

4. Work to provide support for the reflective claims that were made. For example, the article is co-authored by several people, presumably all involved at some level in the process and representing different stakeholder perspectives. Making clear the extent to which different stakeholder participants concur with the reflective narrative would help to support claims. Were there points of divergence of opinion? The data appearing in Box 1 also provide support for claims. Are there similar data available to be used in this way? Elaborate on how these data were collected.

We have provided more evidence (Stories of Change) and we have included the source of information.

Regarding the divergence of opinions what was common was for several suggestions to be made about how to resolve a problem and eventually a decision would be made to choose the most feasible. So there were no major controversies we therefore have not provided more data on this.

- 5. Provide a thoughtful section on implications for M and E practice as well as an agenda for ongoing inquiry in this area. In doing so the contribution of the study will become much more apparent.**

We had provided the key implications for M and E practise in our conclusions and still see these as the major implications for M and E from our work. We have however proposed suggestions for further inquiry as suggested.

Reviewer 2: Katharine Shelley

- 2. More elaboration on capacity building is required given the thrust of the paper is on participatory M&E. For example, did the project team endeavour to strengthen the capacity of any local or district-level stakeholders to continue these participatory M&E approaches once the 3-year implementation period was over? If not, then more details should be provided about why capacity building was either outside the scope of the project and/or too difficult to incorporate. While “capacity building of leaders in management” is one of the key interventions of the MANIFEST project (p.5), the authors have only made brief mention of capacity building (p.17). The paper would be greatly improved by tackling this topic in more detail.**

Thank you very much for this advice. The capacity building approaches that were used in the project included the trainings that were done for various groups of implementers and the learning by doing approach in which the project was implemented with the district stakeholders (district health management team and sub county implementation committee, VHT's, community development officers) taking leading roles in implementation of the study. We have added this in the section where we describe the MANIFEST project. More details of this are also available in the design paper to which we have referred the readers. See lines 21- 23 (page 7/8)

- 3. Data triangulation is briefly mentioned, and supported by evidence, at the beginning of the discussion. Subsequently, the authors suggest triangulation is THE key strength of the paper. If it is the key strength, then I suggest introducing triangulation earlier in the paper, in methods and/or results, to set the reader up for how the various M&E approaches come together to triangulate findings and support decision making. For example, at the quarterly stakeholder meetings were there explicit efforts to triangulate data from**

multiple sources? As currently written, the M&E approaches appear largely standalone pieces of the larger project.

Thank you for this comment however we believe that we had indicated in the methods section that we were collecting data from several sources using several data collection methods. In addition, figure 1 highlights different data collection methods such as surveys, health facility assessment, formal meetings, focus group discussion, key informant interviews and records reviews. In the results section for example under the section on the community we also show how these different methods were used to estimate the actual number of deaths and explain the high number of maternal deaths see lines 4-14 (pg 15), lines 9 page 17– line 7 page 18 and Line 1-11 page 19. We also collected data during support supervisions that were conducted every quarter at each of the health facility and household surveys (page 22).